# An Overview of Mesenchymal Stem Cell Heterogeneity and Concentration

**DOI:** 10.3390/ph17030350

**Published:** 2024-03-07

**Authors:** Elvira Maličev, Katerina Jazbec

**Affiliations:** 1Blood Transfusion Centre of Slovenia, Šlajmerjeva 6, 1000 Ljubljana, Slovenia; katerina.jazbec@ztm.si; 2Biotechnical Faculty, University of Ljubljana, Jamnikarjeva ulica 101, 1000 Ljubljana, Slovenia

**Keywords:** mesenchymal stem cells, heterogeneity, concentration, phenotype, tissue source

## Abstract

Mesenchymal stem cells (MSCs) are of great interest in cell therapies due to the immunomodulatory and other effects they have after autologous or allogeneic transplantation. In most clinical applications, a high number of MSCs is required; therefore, the isolated MSC population must be expanded in the cell culture until the desired number is reached. Analysing freshly isolated MSCs is challenging due to their rareness and heterogeneity, which is noticeable among donors, tissues, and cell subpopulations. Although the phenotype of MSCs in tissue can differ from those of cultured cells, phenotyping and counting are usually performed only after MSC proliferation. As MSC applicability is a developing and growing field, there is a need to implement phenotyping and counting methods for freshly isolated MSCs, especially in new one-step procedures where isolated cells are implanted immediately without cell culturing. Only by analysing harvested cells can we correctly evaluate such studies. This review describes multilevel heterogeneity and concentrations of MSCs and different strategies for phenotype determination and enumeration of freshly isolated MSCs.

## 1. Introduction

Multiple studies have been conducted since 1990 to investigate the regenerative and immune modulatory properties of mesenchymal stem cells (MSCs) [1,2,3,4,5,6]. These studies paved the way for the use of MSCs in preclinical and clinical trials. Over time, the number of applications using MSCs has increased. By 2023, registered clinical trials involving MSCs for treatment at clinicaltrials.gov (accessed on 1 December 2023) numbered over 1500. The most frequently listed studies included cardiology, traumatology, pneumology, neurology, haematology, ophthalmology, and plastic surgery. Studies have shown that MSC therapy is feasible and safe and that MSCs can effectively treat diseases or reduce the side effects of other treatments in some cases and under certain conditions. Studies have demonstrated that the main therapeutic effects of MSCs came from their production and secretion of different bioactive molecules. They can produce immunosuppressive factors such as IDO (indoleamine-2,3-dioxygenase), NO (nitric oxide), IL-10 (interleukin 10), TGF-β (transforming growth factor β), CCL2 (chemokine ligand 2), PGE2 (prostaglandin E2), IL-1Ra (IL-1 receptor antagonist), M-CSF (macrophage colony-stimulating factor), TSG-6 (tumour necrosis growth factor-induced protein 6), SOD-3 (superoxide dismutase 3), and HLA-G5 (human leukocyte antigen-G5) [7,8,9,10,11]. MSCs also secrete many trophic factors with mitogenic, antiapoptotic, neurodegenerative, angiogenetic, and other properties that accelerate the repair of damaged tissues. These factors include EGF (epidermal growth factor), VEGF (vascular endothelial growth factor), hGF (hepatocyte growth factor), PDGF (platelet growth factor), Ang (angiopoietin), and IGF-1 (insulin-like growth factor-1) [12]. The release of trophic factors, which stimulate endogenous mechanisms for regeneration, is essential for their effectiveness. MSCs can also exert antimicrobial activity by secreting IL-37 (human cathelicidin) and lipocalin [13].

Because of their immunomodulatory abilities, MSCs are increasingly gaining recognition not only in human medicine but also in veterinary medicine, where they represent a potential therapeutic option for various diseases, including orthopaedic, orodental, and digestive tract diseases. Human MSCs are under investigation as a treatment for more than twenty clinical conditions. Even though the majority of MSCs die soon after application and engraftment has been found to be limited [14,15,16], their therapeutic usefulness has been shown in many cases. As there are still numerous trials in progress, many researchers believe their results will open up new and more extensive possibilities for their use in future treatments. Commercial products with allogeneic MSCs have already been approved for routine use in the last few years, and this shows us that MSCs are also commercially attractive therapeutic tools.

However, there is still a lack of standardisation in the MSC manufacturing and quality control process. Cell preparation starts with MSC harvesting from a tissue sample. A review of currently registered clinical trials showed that therapeutic MSCs are most frequently obtained from the iliac crest, placenta, or adipose tissue. MSC dosages vary between different therapeutic targets, patients’ conditions, delivery routes, and the types of MSCs used [17]. An analysis of clinical trials with intravenous administrated MSCs between 2004 and 2018, which reported positive outcomes, indicated minimal effective doses ranging from 70 to 190 million MSCs [18]. The MSC count in the management of knee osteoarthritis is slightly lower; in the meta-analysis of results of clinical trials, the group of patients with 50–100 million MSCs delivered to the target site showed significant improvement in pain and functional outcomes, while a higher dose was less effective [19]. Multiple administrations of MSCs with up to several hundred million cells are needed for certain patients and diseases to achieve the desired therapeutic effect [20,21]. To reach such high numbers, MSCs must be expanded in the cell culture. Isolated cells are usually placed in a cell culture without prior determination of their number. MSCs are counted only later, during their proliferation in cell culture, or at the end of the manufacturing process when the final cell product is prepared for application.

Although it is still not routinely performed, MSC counting immediately after isolation from harvested tissue has value. Determining the starting cell number allows us to better plan the cell production process and reduce costs. In Europe and the USA, expanded MSCs are classified as advanced therapy medicinal products (ATMPs). While this regulation ensures quality products, it is also associated with higher costs. For some clinical conditions, using MSCs without multiplying them is becoming increasingly attractive because such minimal manipulation of cells allows MSCs to fall outside the ATMP classification, making the procedure more widely applicable. However, in these newer procedures, freshly isolated cell counting becomes essential because the cell application is performed on the same day within a hospital setting. An example of such one-step procedures involves using autologous bone marrow cells, which take approximately two hours from bone marrow aspiration to cell implantation [12].

Regardless of whether MSCs are used after proliferation or immediately after being obtained from tissue, determination of the initial MSC count is not routinely performed. The most probable reason MSC counting is not carried out before cell culturing or before direct re-implantation is that the MSC population in tissue is sparse. For example, MSCs present only 0.001–0.01% of the mononuclear cells in bone marrow [2]. The second reason is their phenotypic complexity (Figure 1). MSCs cannot be detected based on single marker protein expression; their identification is based on the presence of some specific surface markers and, simultaneously, on the absence of others. Besides this, MSCs show considerable phenotypic differences depending on their source [17]. The heterogeneity of MSCs is also seen in the tissue itself and is reflected in various MSC subpopulations. Furthermore, changes in MSC phenotype can be detected during cell expansion in vitro [22].

In this review, we discuss MSC multilevel phenotypic diversity. As a result of their heterogenicity coupled with their low numbers, MSC phenotypisation and further number determination is not routinely performed before cell expansion. We also summarise all currently available phenotypisation and quantification methodologies for freshly isolated MSCs, although these procedures are not standardised and used regularly, as in the case of multiplied cells.

## 2. Mesenchymal Stem Cell Concentrations in the Source Tissues

Preparing cells for transplantation starts with cell isolation from the tissue source. Because of the small number of MSCs, isolated cells usually do not qualify for transplantation and have to be expanded in vitro over several population doublings to obtain enough cells for implantation. MSCs were first isolated from bone marrow [23], and later, they were obtained from almost every tissue of the body, i.e., bone marrow, adipose tissue, umbilical cord blood, Wharton’s jelly, amniotic fluid, neurospheres, placenta, dental pulp, peripheral blood, synovium and synovial fluid, endometrium, compact bone, dermis, pancreatic islets, adult brain tissues, skeletal muscle tissues, the lungs, heart, and hair follicles. Bone marrow aspiration is an invasive procedure, but it is still a frequently used method of obtaining MSCs for cell therapy. Bone marrow is the most studied source of MSCs and represents the most common source of MSCs for clinical applications in orthopaedics [24,25]. Bone marrow-derived MSCs are also used for acute myocardial infarction and ischaemic heart failure [26,27,28]. The separation of MSCs from peripheral blood after their mobilisation from bone marrow represents an alternative method of cell collection [29]. From 25 mL of bone marrow placed in an adherent cell culture, 100–150 million MSCs can be produced in a few weeks [30]. MSC expansion from the umbilical cord and peripheral blood is performed similarly [31].

Methods of isolating adipose-derived MSCs rely on the digestion of the adipose tissue using enzymes, such as collagenase, which destroys the extracellular matrix [32,33]. Whereas cutting, harvesting, and centrifuging are considered to be minimal manipulations of the cells, enzymatic digestion is a substantial manipulation. Enzymes can cause changes in cell characteristics, and MSCs obtained this way are classified as advanced therapy medicinal products (ATMPs). Some try to achieve cell isolation by optimizing various procedures to avoid enzymatic digestion by concentrating the lipoaspirate or isolating the stem cells with the use of mechanical procedures [34]. In general, these isolation procedures are still not as effective as those with the help of enzymes. One such method for isolating MSCs is the explant culture method, where tissue from Wharton’s jelly, bone, or cornea is cut into smaller pieces and placed in culture dishes [35]. Other methods are immuno-magnetic cell separation and the cell sorting methodology; however, these methods require MSC identification before cell isolation and are primarily used for research purposes. Identification or phenotypisation is also needed whenever we want to count isolated cells. MSCs derived from different anatomical tissues may have different developmental origins and different properties and functions, which can complicate their identification.

Cell amounts vary between tissues, but relatively low MSC concentrations are a feature of all MSC deposits in the body. The number of MSCs in tissue may depend on many factors. The concentration of MSCs in bone marrow depends on the patient’s age, medicaments used, condition of health, and bone remodelling diseases [36]. Studies examining the relationship between age and cell concentration in tissues have produced varying results [12,37,38,39,40]. Still, the majority suggest that the number of MSCs decreases with age. This decline in MSC numbers has been observed in bone marrow MSCs and adipose tissue-derived MSCs [12,39,40]. Cell growth was also more vigorous when cells were isolated from younger rather than older donors [41]. This is consistent with the observation that the ability of MSCs to regenerate tissue is probably impaired by the loss of stem cell numbers and function with age [42].

In the case of bone marrow, it has been proven that the initial number of stem cells also depends on the aspiration technique used to collect MSCs [12,43]. Stem cells are located in different niches in the medullary cavity [44], and different aspiration techniques may be more or less effective in preferentially extracting cells from various niches. Moreover, it has been demonstrated that the number of harvested cells significantly decreases during repeated aspirations. Table 1 shows data about native MSC numbers, which have been determined from the tissues most commonly used as a source of cells for therapies.

## 3. Heterogeneity of Mesenchymal Stem Cells

MSCs exhibit heterogeneity on multiple levels (Figure 1). This heterogeneity is noticeable among donors, tissues, and cell populations. Studies have shown that age, sex, and physiological status could result in functional differences in MSCs derived from the same tissue of origin of different donors [63,64]. Studies have also compared bone marrow MSCs from various donors and found significant differences in cell growth rates and alkaline phosphatase enzyme activity [65]. The variability in the chondrogenic, osteogenic, or endothelial differentiation from different donors was shown as well [64,66,67].

Another level of heterogeneity among MSCs is an outcome of variability between different tissues. Since the microenvironment has a clear and substantial impact on the cells, heterogeneity is inevitable even when the cells are genetically identical. It has already been shown that the proliferation or differentiation of MSCs depends on the cell source. Studies also report different immunomodulatory properties of MSCs harvested from different tissues [68]. MSC heterogeneity is also present within the same tissue. Heterogeneity in a particular tissue is manifested in different subpopulations with distinct expression profiles and functional properties [69]. MSC plating density, different components in growth media (serum, growth factor combinations), and oxygen tension may affect the mesenchymal population’s gene profile, epigenomic state, and phenotype. Clones derived from even single proliferating MSCs can be heterogenous in vitro [70]. Additionally, clonal analysis of expanded MSCs has shown that the diversity is dramatically lowered to just a few clones after multiple passages [71]. As a result, the MSC subpopulations in cell culture do not represent the most abundant clones present at the beginning [71,72,73,74]. Therefore, it must be considered that the expression profile of MSCs in tissue probably differs from that of proliferating MSCs in culture. The phenotype during culturing can be monitored after each cell passage. With a larger number of passages, more MSCs are obtained for clinical application. However, since the number of cell passages can affect the MSCs’ properties, the most commonly used passages for treating include passages 3 to 7.

A consequence of MSC heterogeneity is the complexity of their identification, especially in the tissue. This is further complicated by the fact that MSC subpopulations are not well defined and are difficult to distinguish from each other. Examples of different combinations of phenotype markers used for MSC identification are collected in Table 2. Various marker combinations were used to confirm the identity of MSCs from different sources, i.e., bone marrow, adipose tissue, umbilical cord, and placenta. The most commonly used markers were CD29, CD44, CD73, CD90, and CD105, which were positively expressed, while among negatively expressed markers, CD34, CD45, and HLA-DR were most common.

## 4. Counting Mesenchymal Stem Cells Harvested from Tissue

### 4.1. The Mesenchymal Stem Cell Counts per Total Nucleated Cells

MSC estimation per total nucleated cells is the most simple method for determining native MSC numbers, but it is inaccurate. Hernigou et al. reported an equation for the prediction of the number of nucleated cells in bone marrow, as follows: N(10^8^/kg) = (V × NP) − (V − 100) × NS/P, where V is the total volume of aspirate, NP is the nuclear cell count per ml of bone marrow aspirate, NS is the nuclear cell count per ml of peripheral blood, and P is the patient’s weight [112]. The above calculation was applied to MSCs isolated from bone marrow. Stem and progenitor cells in the uncultured stromal vascular fraction (SVF) from adipose tissue usually amount to up to 3% of the whole number of cells, which is 2500-fold greater than the frequency of stem cells in bone marrow [113].

### 4.2. Estimation of Mesenchymal Stem Cell Count with the CFU-F Test

MSCs were initially referred to as fibroblast-like cells that can generate colony-forming unit fibroblasts or CFU-F. Determining the colony-forming units present is the oldest way of estimating the actual stem cells in the suspension of different cells [114]. A suspension of mixed cells is plated on plastic culture dishes at a low density. Culturing under specific conditions allows stem cells to adhere, proliferate, and form colonies. After the growth of visible colonies, they are counted using light microscopy. The number of colonies indicates the number of cells that were able to proliferate, as each colony is presumed to originate from a single stem cell.

The CFU-F assay has some shortcomings. Cell culturing takes several days before the identification of colonies; the method is time-consuming and inappropriate for routine use, the counting of colonies is subjective, not standardised; and the technique lacks accuracy [115]. Even though this procedure gives us the number of functional, proliferating MSCs in a tissue, there is a need for further investigation to explore alternative approaches to assessing the quantity of MSCs.

## 5. Mesenchymal Stem Cell Quantification with Immunophenotyping

For accurate counting of MSCs, their identity has to be ascertained first. Identification is based on the characteristic protein expression profile of MSCs. The Mesenchymal and Tissue Stem Cell Committee of the International Society for Cellular Therapy (ISCT) defined an MSC phenotype that is used irrespective of tissue source. Expanded cells are determined to be MSCs when expressing CD105, CD90, and CD73 and not expressing CD45, CD34, CD14 (CD11b), CD79 (CD19), or human lymphocyte antigen-DR (HLA-DR) [75]. This method is usually not used in the case of freshly isolated MSCs. Besides this, in recent years, new markers with positive or negative expression and new marker combinations have been recognised, among them CD49a, CD106 (VCAM-1), CD140b, CD146, CD271 (LNGFR), MSCA-1, GD2, and STRO-1 [12,79,116,117,118]. Many of these markers were used to identify MSC subpopulations expressing different types of regulatory proteins that function in hematopoiesis, angiogenesis, neural activities, and immunity processes [63].

In order to quantify MSCs before culturing, it is necessary to identify the MSC (sub)population from the harvested cell suspension where MSCs are still suspended with other cells. MSCs from the same tissue are heterogeneous and may not share the same phenotypical markers. Additionally, there is a lack of consensus on the markers identifying or distinguishing MSCs derived from different tissues. The chosen phenotype markers may influence the count; for example, the CD45^–^/CD271^+^ MSCs correlate more with CFU-F numbers than the CD45^–^/CD73^+^/CD90^+^/CD105^+^ MSCs [12]. Therefore, the identification of native MSCs is much more complicated than expanded MSCs. The biological capacity of MSCs (i.e., immunomodulatory capacity, differentiation potential to a specific cell type, and endogenous stem cell mobilising capacity) of one tissue source may differ from others. Isolation or production of a specific subpopulation of MSCs for a particular medical indication should improve outcomes.

Despite the numerous markers for MSC identification at our disposal, no agreement exists over the correct composite marker panel for defining MSCs in freshly isolated and non-cultured populations. For this reason, a precise characterisation of MSCs derived from different tissues and their proper ties relating to their therapeutic potential represents an essential requirement for the exploitation and development of optimal MSC-based therapies. Additional studies should be conducted to test the correlation between phenotype markers and CFU-F and other biological effects. Together with multicentric study data, the ISCT could recommend criteria for native MSCs analysis.

## 6. Towards the Standardisation of Mesenchymal Stem Cell-Based Therapies

Knowing the number of MSCs before transplantation would help clarify the efficacy and limitations of cell therapy. When planning for autologous transplantations, it is important to have information on the initial number of cells in order to anticipate cell proliferation and determine the date of application (freezing, thawing, and additional culturing can be avoided). In cases where one-step procedures without cell expansion are used, it would be beneficial to determine the count and subpopulations of MSCs present for further evaluation of the treatment. By detecting MSC properties and sorting selected subpopulations, it may be possible to utilise the most efficient cells for a given clinical condition.

There are many groups that are currently working on standardising therapies that involve the use of MSCs. Their focus is on standardising MSC isolation techniques, enzymatic digestion, explant culture, or enrichment techniques. They are also discussing nomenclature and trying to set identifying criteria to distinguish MSCs from different tissue cells and other fibroblastic cells. The ISCT has recently published ISO standardisation documents that are focused on the biobanking of MSCs from two tissue sources, Wharton’s Jelly and bone marrow [119,120,121,122]. There is an awareness that these documents are evolving documents with iterations amending various sections with revisions as the science progresses and we gain deeper understanding of MSC biology and functionality.

## 7. Conclusions

There is still a lack of standardisation in the MSC manufacturing and quality control process; however, there are many groups that are currently working on standardising therapies that involve the use of MSCs with a recognition that premature standards or inappropriate scope may distort or inhibit the adoption of MSC-based therapies. Currently, there is some information available on cultured cells, but more research is needed on native MSCs, especially those used for one-step procedures. Uncultured MSCs offer an alternative in some clinical applications. The main reason that initial MSCs are not counted is their low concentration in native tissue and their multilevel heterogeneity. Consequently, the identification of native MSCs is more complicated than that of cultured cells. Since the identification of MSCs with cell surface markers is necessary for their counting, there is a need to develop a consensus regarding standards for more accurately identifying native MSCs isolated from different sources and different MSC subpopulations.

## Figures and Tables

**Figure 1 pharmaceuticals-17-00350-f001:**
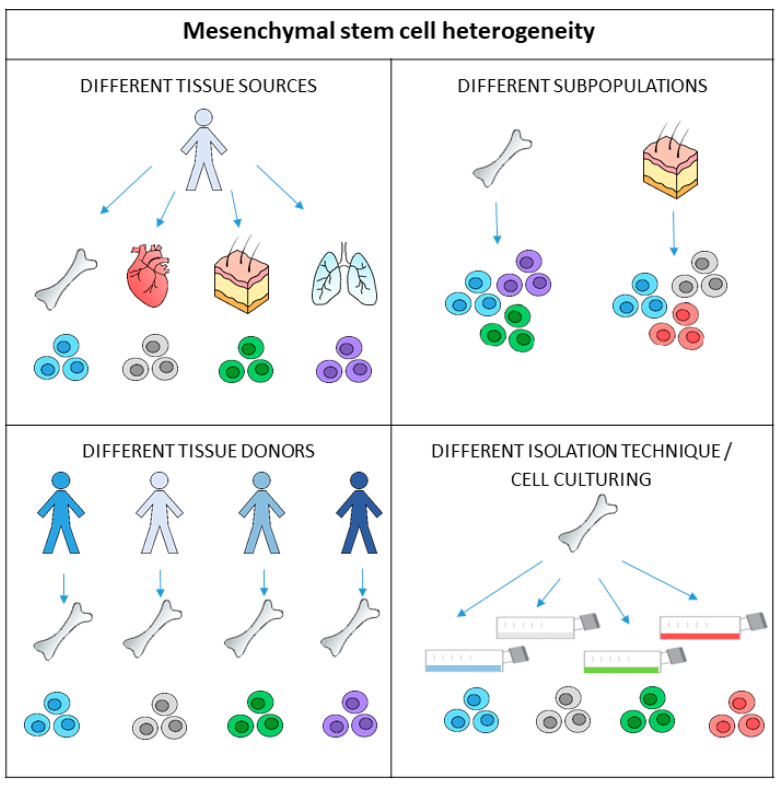
Mesenchymal stem cells (MSCs) exhibit heterogeneity on multiple levels. The different biological properties and functions of MSCs are due to the anatomical tissue sources they come from as the environment affects the genetically identical cells. Differences can also be observed in MSCs taken from the same tissue sources in different individuals. There are also different subpopulations of MSCs present in tissues, and their proliferation, differentiation, or immunomodulation functions may differ. Furthermore, isolation and cell culturing techniques may affect the phenotype and functional properties of cells before transplantation.

**Table 1 pharmaceuticals-17-00350-t001:** Amount of mesenchymal stem cells (MSCs) in different tissue sources.

MSC Source	Amount of MSCs (Concentration/Percentage/Number of CFU-F)	Reference(s)
Bone marrow	The number of fibroblast colonies for 5 × 10^6^ bone marrow cells was 68 ± 10 (× ± 1 SEM), and the range was 45–143	[45]
	Some 0.01% to 0.001% of mononuclear cells isolated on a density gradient (Ficoll/Percoll) give rise to plastic adherent fibroblast-like colonies	[2]
	Less than 1 MSC in 100,000–500,000 nucleated cells in bone marrow aspirate from adults	[46]
	In bone marrow aspirate: 0.018% CD45^−^/CD73^+^/CD90^+^/CD105^+^ cells	[12]
	In bone marrow: 0.001% CD45^−^/CD11b^−^/CD19^−^/CD34^−^/HLA-DR^−^/CD73^+^/CD90^+^/CD105^+^	[47]
	In bone marrow: 0.026% CD45^–/low^ CD271^bright^ cells	[48]
	In bone marrow: 0.007% (chip cytometry) and 0.014% (flow cytometry)	[49]
	In bone marrow: 1 MSC per 10^5^ adherent stromal cells	[50,51]
	3 × 10^5^ adherent cells/g of bone marrow	[52]
Adipose tissue	2 × 10^5^ adherent cells/g of adipose tissue	[52]
	The yield of stromal vascular cells for the abdomen was 0.7 × 10^6^ ± 0.1 × 10^6^ cells/g adipose, the hip and thigh region 0.5 × 10^6^ ± 0.07 × 10^6^ cells/g, and the mamma 0.6 × 10^6^ ± 0.3 × 10^6^ cells/g	[53]
	5 × 10^5^ stem cells could be obtained from 400 to 600 mg of adipose tissue	[54]
	Approx. 0.5 × 10^4^ to 2 × 10^5^ stem cells can be isolated per gram of adipose tissue	[55]
	2–6 × 10^8^ processed lipo-aspirate cells can be obtained from 300 mL adipose tissue	[56]
Umbilical cord blood	The frequency of MSC-like cells ranged from 0 to 2.3 clones per 1 × 10^8^ mononuclear cells	[57]
	From 1 × 10^3^ to 5 × 10^3^ cells per sample	[58]
Umbilical cord stroma	4 × 10^5^ cells per sample; 10–15 × 10^3^ cells per centimetre of cord	[59]
	From 15–17 × 10^3^ cells/cm of umbilical cord length, with a range of 10–50 × 10^3^ cells/cm	[60]
Placenta	0.6% to 2.1%	[61,62]

**Table 2 pharmaceuticals-17-00350-t002:** An overview of different combinations of phenotype markers used for MSC identification.

MSC Source	Positive Expression	Negative Expression	Reference(s)
Bone marrow	CD73, CD90, CD105 (≥95% positive)	CD11b or CD14, CD19 or CD79a, CD34, CD45, HLA-DR (≤2% positive)	[75]
	CD105, CD73, CD90, CD44, CD49d, CD49f, PDGFRβ	CD45, CD34, CD19, CD14, HLA-DR, carcinoma cell marker (c-MET), EPCAM	[76]
	STRO-1		[77,78]
	Melanoma-associated cell adhesion molecule (MCAM/CD146)		[79]
	CD271		[80,81]
	CD105, LNGFR, HLA-DR, CD10, CD13, CD90, STRO-1, bone morphogenetic protein receptor type IA (BMPRIA)	CD14, CD34, CD117, CD133	[82]
	CD10, CD73, CD140b, CD146, GD2, CD271.		[83]
	CD73, CD90, CD105 (>90%)	CD34, CD45, HLA-DR (<5%)	[84]
	CD90, CD105, CD44, CD10, CD271, HLA-ABC		[85]
	CD105, CD44, CD29, CD90, and CD106	CD14, CD31, CD34, CD45	[86]
	CD9, CD10, CD13, CD73, CD105, CD166, frizzled-9 (FZD-9), W8B2 (anti-MSCA-1)		[87]
	PDGFRα, CD51		[88]
	CD271, CD10, CD13, CD73, and CD105, W3D5, W5C5 (SUSD2), W8B2		[89]
	CD13, CD15, CD73, CD140b, CD144, CD146, CD16		[90]
	Cellular prion protein (PrP)		[91]
	CD29, CD44, CD105, CD26, CMKLR1	CD14, CD34, CD45, HLA-DR	[92]
	CD29, CD44, CD54, CD73, CD90, CD105, Nestin, SOX2, vascular cell adhesion molecule 1 (VCAM/CD106)	CD11b, CD19, CD31, CD34, CD45, HLA-DR	[93]
	CD44, CD63, CD73, CD105, melanoma cell adhesion molecule (MCAM/CD146)	CD14, CD34	[94]
	CD44, CD90, CD105, CD166, CD146	CD31, CD34, CD45, HLA-DR	[95]
Adipose tissue	CD29, CD44, CD54, CD73, CD90, CD105, Nestin, SOX2	CD11b, CD19, CD31, CD34, CD45, HLA-DR, VCAM/CD106	[93]
	CD90, CD105, CD73, CD44	CD11b, CD19, CD34, CD45, HLA-DR	[96]
	CD90, CD44, CD29, CD105, CD13, CD34, CD73, CD166, CD10, CD49e, CD59; HLA-ABC, STRO-1	CD31, CD45, CD14, CD11b, CD34, CD19, CD56, CD146; HLA-DR	[97]
	CD34, CD90		[98]
	CD29, CD34, CD44, CD49d, CD73, CD90, CD105, and CD151	CD45	[33]
	CD34, CD90, CD73, CD105, CD44	CD235a, CD45, CD31	[99]
	CD73, CD90, CD105, CD271	CD34, CD45	[100]
	CD271		[101]
	CD44, CD90, CD105, CD166	CD31, CD34, CD45, CD146, HLA-DR	[95]
Umbilical cord	CD29, CD44, CD54, CD73, CD90, CD105, Nestin, SOX2, VCAM/CD106	CD11b, CD19, CD31, CD34, CD45, HLA-DR	[93]
	CD105 (SH2), CD73 (SH3), CD90 (Thy-1), CD44	CD34, CD45	[102]
	CD105, CD90, CD73, CD54, CD13, CD29, CD44	CD31, CD14, CD34, CD45	[103]
	Stro-1, CD44, CD105, CD146		[104]
	CD13, CD29, CD44, CD73, CD90, CD105, CD146, CD166, HLA-ABC	CD14, CD34, CD45, CD117, CD133, CD144, CD326, HLA-DR	[105]
	CD13, CD29 (integrin β1), CD73 (SH3), CD90 (Thy-1), CD105 (SH2), HLA-ABC	CD34, CD45, CD133, HLA-DR	[106]
	CD73, CD90, CD105	CD34, CD45, and HLA-DR	[107]
	CD13, CD29, CD44, CD90, CD105, CD146	CD10, CD14, CD34, CD117	[108]
	CD73, CD90, CD105, CD166, SOX2, SSEA4	CD14, CD34, CD45, HLA-DR, CD106	[109]
Placenta	CD9, CD10, CD13, CD73, CD105, CD166, stage-specific embryonic antigen-4 (SSEA-4), FZD-9		[87]
	CD29, CD44, CD54	CD31, CD45, AC133	[110]
	CD13, CD73, CD90, CD105, CD146, CD140b, HLA-ABC	CD14, CD34, CD45, CD66b, CD324, CD326, HLA-DR	[111]
	CD73, CD90, CD105, CD166, SOX2, SSEA4, CD106	CD14, CD34, CD45, HLA-DR, CD106	[109]

## Data Availability

Data sharing is not applicable.

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
