# Peer review of "An Overview of Mesenchymal Stem Cell Heterogeneity and Concentration"

_pharmaceuticals, 2024, doi:10.3390/ph17030350_

Round 1
Reviewer 1 Report
Comments and Suggestions for Authors
Dear Authors,
I've done a review of your manuscript and would like to provide my opinion.
Authors focused on the problem with the lack of standarization of MSC-based therapies that do not require cell expansion - there is lack of standardized and accurate methods to determine the initial number of MSCs in the isolate obtained from source tissue. The lack of knowledge on the concentration and phenotype of administered cells can be a reason of various therapeutic effects observed between the patients.
Authors identified some methods commonly used to determine the initial amount of cells in cell isolate, however none of them could be employed as standard to facilitate standardization of MSC therapies due to their limitations.
However, there is no information of the current trends and studies focused on standardization of such therapies (many many groups work on this), please see:
- https://academic.oup.com/stcltm/article/12/11/745/7274806
- https://www.frontiersin.org/articles/10.3389/fcell.2021.632717/full
- https://www.ncbi.nlm.nih.gov/pmc/articles/PMC8109215/
- https://www.isct-cytotherapy.org/article/S1465-3249(23)00100-7/fulltext
In my opinion the literature review is incomplete, as plenty of important papers were not included within the manuscript. Moreover, it would be the most interesting to know which direction should be taken to develop methods that will allow to determine the initial level of MSCs in freshly obtained isolates.
Lastly, the writing style should be strongly improved, possibly by native speaker as many of sentences are hard to understand by their style, there are many of colloquialisms as well that should be avoided in the scientific literature (please see comments).

The writing style and language should be strongly improved, possibly by native speaker as many of sentences are hard to understand by their style, there are many of colloquialisms as well that should be avoided in the scientific literature (please see comments).
Author Response
Revised manuscript and response to reviewer are attached.

Reviewer 2 Report
Comments and Suggestions for Authors
Minor concerns:
· Line 97-98: Currently, MSCs are mostly grown to ---------adult treatment. I think so, the authors should mention the passage number (P-No.) with the number of MSC.
· Line 139. Bone marrow space can be written as the medullary cavity.
· Line 154: “Sex” should be replaced by gender.
· Line 255: If MSCs are used for therapy (can be replaced by therapeutic purposes)
Major concerns:
· Figure 1. I could not understand the figure 1. This needs to be rearranged or elaborated.
· The authors did not mention any passage number because whenever we need to culture MSCs for therapeutic purposes, we need a large number of cells, and different passage numbers will certainly be used. The author should write (4.3) on the paragraph number.
Author Response

(The authors gave the same response as above.)

Round 2
Reviewer 1 Report
Comments and Suggestions for Authors
Dear Authors,
thank you for giving me the opportunity to help in improving your manuscript and for considering my comments as relevant.
The document was upgraded and questions that occurred answered. I am happy that MSC field is developing.
Best wishes
Reviewer 2 Report
Comments and Suggestions for Authors
NA